# BTEAC Catalyzed Ultrasonic-Assisted Synthesis of Bromobenzofuran-Oxadiazoles: Unravelling Anti-HepG-2 Cancer Therapeutic Potential through In Vitro and In Silico Studies

**DOI:** 10.3390/ijms24033008

**Published:** 2023-02-03

**Authors:** Ali Irfan, Ameer Fawad Zahoor, Azhar Rasul, Sami A. Al-Hussain, Shah Faisal, Sajjad Ahmad, Rida Noor, Muhammed Tilahun Muhammed, Magdi E. A. Zaki

**Affiliations:** 1Department of Chemistry, Government College University Faisalabad, Faisalabad 38000, Pakistan; 2Department of Zoology, Government College University Faisalabad, Faisalabad 38000, Pakistan; 3Department of Chemistry, College of Science, Imam Mohammad Ibn Saud Islamic University (IMSIU), Riyadh 11623, Saudi Arabia; 4Department of Chemistry, Islamia College University Peshawar, Peshawar 25120, Pakistan; 5Department of Health and Biological Sciences, Abasyn University, Peshawar 25000, Pakistan; 6Department of Pharmaceutical Chemistry, Faculty of Pharmacy, Suleyman Demirel University, Isparta 32260, Turkey

**Keywords:** bromobenzofuran-oxadiazole, BTEAC, HepG-2 cell line, EGFR, PI3K, mTOR and tubulin polymerization inhibitors, SAR, molecular docking, MD simulations, DFT studies

## Abstract

In this work, BTEAC (benzyl triethylammonium chloride) was employed as a phase transfer catalyst in an improved synthesis (up to 88% yield) of S-alkylated bromobenzofuran-oxadiazole scaffolds **BF1-9**. These bromobenzofuran-oxadiazole structural hybrids **BF1-9** were evaluated in vitro against anti-hepatocellular cancer (HepG2) cell line as well as for their in silico therapeutic potential against six key cancer targets, such as EGFR, PI3K, mTOR, GSK-3β, AKT, and Tubulin polymerization enzymes. Bromobenzofuran structural motifs **BF-2**, **BF-5**, and **BF-6** displayed the best anti-cancer potential and with the least cell viabilities (12.72 ± 2.23%, 10.41 ± 0.66%, and 13.08 ± 1.08%), respectively, against HepG2 liver cancer cell line, and they also showed excellent molecular docking scores against EGFR, PI3K, mTOR, and Tubulin polymerization enzymes, which are major cancer targets. Bromobenzofuran-oxadiazoles **BF-2**, **BF-5**, and **BF-6** displayed excellent binding affinities with the active sites of EGFR, PI3K, mTOR, and Tubulin polymerization enzymes in the molecular docking studies as well as in MMGBSA and MM-PBSA studies. The stable bindings of these structural hybrids **BF-2**, **BF-5,** and **BF-6** with the enzyme targets EGFR and PI3K were further confirmed by molecular dynamic simulations. These investigations revealed that 2,5-dimethoxy-based bromobenzofuran-oxadiazole **BF-5** (10.41 ± 0.66% cell viability) exhibited excellent cytotoxic therapeutic efficacy. Moreover, computational studies also suggested that the EGFR, PI3K, mTOR, and Tubulin polymerization enzymes were the probable targets of this **BF-5** scaffold. In silico approaches, such as molecular docking, molecular dynamics simulations, and DFT studies, displayed excellent association with the experimental biological data of bromobenzofuran-oxadiazoles **BF1-9**. Thus, in silico and in vitro results anticipate that the synthesized bromobenzofuran-oxadiazole hybrid **BF-5** possesses prominent anti-liver cancer inhibitory effects and can be used as lead for further investigation for anti-HepG2 liver cancer therapy.

## 1. Introduction

Cancer is a serious threat to today’s world and is considered the second foremost cause of death globally. The search for new anticancer therapeutic agents has gained considerable attention and interest. Many research groups of synthetic/medicinal chemists and pharmacologists are working to discover and develop novel anticancer agents for the treatment of around 100 different types of cancers [1,2,3,4]. Liver cancer is the fourth most common cause of deaths among all the cancers, and it is the seventh leading cause of death in women and the fifth in men worldwide. According to the statistics, liver disease will become a global problem by 2025, affecting almost one million people across the world. The major cause of hepatocellular carcinoma (HCC) is obesity, hepatitis C and B viruses (HCV& HBV), diabetes, and alcoholic and nonalcoholic fatty liver diseases (AFLD & NAFLD) [5,6,7,8,9]. Various treatments are in practice to treat cancerous cells; 49% of cancer cells are treated with surgery, 40% with radiotherapy, and only 11% are being cured with chemotherapy.

Various chemotherapeutics are available to cure different cancers, yet most of them suffer from certain drawbacks (being expensive, often having less efficacy than required, and carrying a lot of negative side effects) [10,11,12,13]. The development of new bioactive privileged structural motifs based on molecular recognition is the core and significant objective of medicinal chemistry. The oxygen and nitrogen containing heterocycles have always been of great synthetic/biological interest owing to their broad spectrum of applications in the fields of medicinal chemistry, pharmaceutics, and pharmacology. The compound’s structures innately lock the therapeutic potential against various diseases and diversified structures of heterocyclic scaffolds are the main dynamic source to act as biologically active therapeutics. Therapeutically, the heterocyclic oxadiazole is the one of the most famous pharmacophores of oxygen and nitrogen, containing a five member azole family that is an integral structural unit of various clinical drugs, as displayed in Figure 1. In general, the biological properties manifested by oxadiazole moiety bearing molecules include anti-fungal, anti-inflammatory, analgesic, anti-diabetic, anticancer, antibacterial, antiviral, antibacterial, antitumor, hypotensive, anti-obesity, etc. [14,15,16,17,18,19].

As per recent literature, benzofuran derivatives also exhibit a wide spectrum of versatile biological activities, such as a bone anabolic agent, anti-inflammatory, antimicrobial agents, monoamine oxidase (MAO) inhibitors, tumor necrosis factor-α (TNF-α) inhibitors, anti-HIV/HCV, CNS related disorders, antipyretic, renal disorders, anticoagulant, anti-lung cancer, dual active ligands of 5-HT1A and serotonin reuptake inhibitors, anti-TB, farnesyltransferase (FTase) inhibitors, aromatase (cytochrome P450) inhibitors, Pim-1 serine/threonine-protein kinase inhibitors, hypoxia-inducible factor (HIF-1α) inhibitors, serotonin receptor inhibitors (5-HT1A), glycogen synthase kinase (GSK-3β) inhibitors, mammalian target of the rapamycin (mTOR) inhibitors, prolyl endopeptidase inhibitors (PEP), cholinesterase activity inhibitors, α-glucosidase inhibitors, tubulin polymerization inhibitors, anti-human liver carcinoma cell line (HepG2), serine-threonine kinase (AKT) inhibitors, PLK1 PBD inhibitors, PI3K/Akt inhibitors and VEGFR-2 tyrosine kinase inhibitors, etc. Some of the most famous benzofuran core based clinically active therapeutics have been depicted in Figure 2 [20,21,22,23,24,25,26,27].

### The Rationale of Molecular Design of Bromobenzofuran-oxadiazoles **BF1-9** against HepG2 Cancer Cell Line

The versatile therapeutic potential of oxadiazoles has made this moiety an important pharmacological scaffold for drug design, especially in the field of oncology [28,29]. A large library of oxadiazoles, such as naproxen-, benzoxazole-, piperazinyl-, thienyl-, and naphthalene-based oxadiazoles and 2,5-disubstituted-1,3,4-oxadiazoles, etc. displayed excellent anti-cancer potential against HepG2 cell line (Figure 3). Benzofuran analogues, such as thiazole-based benzofurans, have been found to be more active scaffolds than the standard drug 5-fluorouracil. Furan-thiazole based oxadiazoles displayed comparable IC_50_ values to the reference drug doxorubicin. Similarly, trifluoro containing benzofuran hybrids exhibited strong anti-proliferative efficacy against HepG-2 cell line as displayed in Figure 3 [24,30,31,32,33,34].

The rationale of the current study was based on the molecular hybridization of bromo-arylated fragment with the combination of two major bioactive heterocyclic moieties furan and oxadiazole, each of which possess excellent and privileged pharmaceutical and pharmacological profiles, as demonstrated in Figure 1, Figure 2 and Figure 3, respectively. In the recent research work, we screened the synthesized substituted *S*-arylatedbenzofuran-oxadiazole scaffolds against the HepG-2 cancer cell line. 

Based on the plethora of the research cited above, we were interested to assess the therapeutic potential of bromobenzofuran-oxadiazoles **BF1-9** as anti-hepatic cancer agents.

## 2. Materials and Methods

### 2.1. Materials for the Synthesis Bromobenzofuran-Oxadiazole Derivatives **BF1-9**

The ultrasonic irradiated experimental synthetic strategy was performed in a 1.9-L capacity ultrasonic cleaner bath (model 1510) powered by a 115 V heater switch, 47 kHz, and mechanical timer. In this study, all analytical grade starting materials, reagents, and solvents were purchased from Alfa Aesar or Sigma Aldrich. The reactions were monitored by thin-layer chromatography (TLC) using aluminum-backed silica gel plates. 

### 2.2. BTEAC Catalyzed Synthesis of Bromobenzofuran-Oxadiazole Structural Hybrids **BF1-9** by Ultrasonic Irradiated Synthetic Approach

A reaction mixture containing bromobenzofuran-oxadiazole-2-thiol **1** (1 mmol) [35,36], BTEAC (10 mol%), and substituted bromoacetanilide derivatives **2a–i** (1.2 mmol) [35,36] in CH_3_CN was sonicated for 30 min at 40 °C (Figure 1). Upon reaction completion, n-hexane was added to afford *S*-arylated bromobenzofuran-oxadiazole derivatives **BF1-9** as precipitates, which were further purified with an ethanolic recrystallization process or column chromatography technique using ethyl acetate–petroleum ether (1:9) [37,38,39].

### 2.3. Anti-hepatocellular Carcinoma MTT Assay

The anti-hepatocellular therapeutic potential of the afforded bromobenzofuran-oxadiazoles **BF1-9** was evaluated by applying the MTT assay, and these structural hybrids **BF1-9** were screened against the HepG2 cancer cell line [40,41]. The Dulbecco’s modified Eagle’s medium was used to develop the HepG2 cell line as a monolayer culture composed of 10% FBS, 100 μg/mL penicillin, and 1% streptomycin. The humid atmosphere at 37 °C was provided for incubation, which consisted of 95% air, 5% CO_2_, and water. The synthesized bromobenzofuran-oxadiazoles **BF1-9** (100 μg/100 mL) were dissolved in DMSO to treat the HepG2 cancer cell line and the DMSO-treated cells were used as a negative control. The 96-well plates were used to culture HepG2 cells overnight, which were further treated with the **BF1-9** compounds.

After the incubation of 48 h, 10 μL, 5 mg/mL of MTT reagent was added to each plate and incubated for 4 h at 37 °C. The percentage cell viability was calculated by measuring the absorbance at 490 nm after the addition of 150 μL DMSO to each plate via a micro-plate reader.

### 2.4. Computational Approach of Bromobenzofuran-Oxadiazoles **BF1-9**

#### 2.4.1. Retrieval of EGFR, PI3K, mTOR, AKT, Tubulin Polymerization, and GSK-3β Protein PDB Structures

For the computational investigations, the target enzymes EGFR, PI3K, mTOR, AKT, Tubulin polymerization and GSK-3β protein PDB structures were retrieved from the RCSB with PDB identifiers 4HJO, 3ML9, and 6Y9R [42,43,44,45] and 3QKL, 4JSX, and 4O2B [46,47,48].

#### 2.4.2. Designing of Ligands and Molecular Docking of Bromobenzofuran-Oxadiazoles **BF1-9**

The Molecular Operating Environment (MOE) 2009.10 was used to carry out the molecular docking analyses. The Biovia DS software was used to prepare the EGFR, PI3K, mTOR, AKT, Tubulin polymerization, and GSK-3β enzyme protein structures for docking experiments by removing any extra water molecules and heteroatoms from their protein PDB structures. The chemdraw professional program was used to prepare the structures of the ligands **BF-2**, **BF-5**, and **BF-6** and saved them in the mol format for later research [49,50,51,52]. The ligand structures were loaded into the MOE before docking, and the MMFF94x forcefield was used to reduce their energy. The protein PDBs was opened in MOE and three d-protonated using the MMFF94x force field. The active site of the six proteins was located and selected using the Site Finder function of MOE. Using the Triangle Matcher placement techniques, the compounds were docked in the binding pocket and scored by the Dock module using the London-dG scoring function of MOE. The ligand-protein interaction was viewed using the Biovia DS Studio software.

#### 2.4.3. ADMET and Drug-Likeness Studies of Bromobenzofuran-Oxadiazoles **BF1-9**

The Swissadme and ADMETlabonline servers were used for the ADME and drug-likeness research, whereas the admetSARonline server was used for toxicity investigations [53,54,55,56].

#### 2.4.4. Molecular Dynamic Simulation of Bromobenzofuran-oxadiazoles **BF-2**, **BF-5,** and **BF-6**

The molecular dynamic simulation was done using AMBER20. Preprocessing of the docked complexes was done via the Antechamber program. The receptors and compounds were processed through FF14SB and GAFF force fields, respectively. The systems were energy minimized for 1500 steps. First, the steepest descent followed by conjugate gradient algorithm was used. Heating was done for 310 K in gradual fashion. The systems were equilibrated and then subjected to production run of 50 ns. The CPPTRAJ module was used for trajectories analysis while XMGRACE was applied for making plots. The MMPBSA.py module was used for binding free energies analysis. In total, 1000 frames were picked for MMPBSA and MMGBSA analyses [57,58,59,60,61,62,63].

#### 2.4.5. DFT Studies of Bromobenzofuran-Oxadiazoles **BF-2**, **BF-5,** and **BF-6**

The DFT computations of biologically active bromobenzofuran-oxadiazoles **BF-2, BF-5** and **BF-6** structural motifs were performed with Gaussian 09. The geometries of compounds **BF-2**, **BF-5**, and **BF-6** were optimized by using the DFT method with B3LYP in the ground state. The basis set was LANL2DZ. Then, energy calculations of **BF-2**, **BF-5**, and **BF-6** were performed by keeping the DFT optimization setups the same. By the time-dependent DFT method, the total energy, the highest occupied molecular orbital (HOMO) energy, and the lowest unoccupied molecular orbital (LUMO) energy were obtained. Thereafter, the related parameters were computed using these values. The DFT computation results were visualized and analyzed using GaussView 5.0 [64,65,66,67]. 

### 2.5. Statistical Data

The statistical data was analyzed with the Prism software, and the results of the study were measured in triplicates and depicted as mean ± SD.

## 3. Results and Discussion

### 3.1. Chemistry

#### Synthesis of Bromobenzofuran-Oxadiazole Structural Hybrids **BF1-9**

In the present research work, BTEAC catalyzed synthetic protocol was applied to furnish substituted 1,3,4-oxadiazole appended bromobenzofuran structural motifs **BF1-9** as sketched in Figure 1. In the methodology, the scaffold **BF-4** containing electron-donating (ED) methoxy group at ortho positions was achieved in maximum 88% yield while the lowest yield was observed for **BF-8** (75%), having highly electronegative electron withdrawing group chlorine at para position (2-position) on the phenyl ring, as shown in Table 1. This BTEAC-catalyzed synthetic approach provided higher reaction yields as compared to our previously reported methodology yields (53–79%) [39].

### 3.2. Biological Evaluation of Bromobenzofuran-Oxadiazoles

#### 3.2.1. Anti-Hepatocellular Carcinoma Activity of Bromobenzofuran-Oxadiazoles **BF1-9**

The anti-HepG2 liver cytotoxicity of bromobenzofuran-oxadiazole structural hybrids **BF1-9** was evaluated via MTT assay. The human liver tumor cell line (HepG2) was used to examine one dose-response, and the results are presented in Table 2. These results highlighted the significance of *S*-alkylated bromobenzofuran-oxadiazole derivatives **BF1-9,** which exhibited significant anticancer activity (Cell viability = 10.41 ± 0.66% to 44.69 ± 6.85%). Among all bromobenzofuran-oxadiazole derivatives **BF1-9**, 3,4-dimethyphenyl-containing scaffold **BF-9** showed the maximum cell viability of 44.69 ± 6.85%, which indicated that this structural hybrid **BF-9** is less cytotoxic while 2-methoxyphenyl containing structural motif **BF-4** (13.88 ± 0.6% cell viability) and 4-fluorophenyl-based derivative **BF-8** (13.85 ± 1.08% cell viability) displayed comparable and better cytotoxic potential with respect to the most bioactive bromobenzofuran-oxadiazoles **BF-2**, **BF-5,** and **BF-6**. The phenyl, 2,5-dimethyl phenyl, and 2-chlorophenyl containing bromobenzofuran-oxadiazoles **BF-1**, **BF-3,** and **BF-7** demonstrated moderate anti-hepatic therapeutic efficacies (26.29 ± 17.54% cell viability, 33.12 ± 6.15% cell viability, and 21.47 ± 8.55% cell viability), respectively, as shown in Table 2. As depicted in Table 2, 3,4-Dichloro-based bromobenzofuran-oxadiazole **BF-2**, (12.72 ± 2.23% cell viability), 2-fluorophenyl containing bromobenzofuran-oxadiazole **BF-6** (13.08 ± 1.08% cell viability), and the least cell viability (10.41 ± 0.66%) and maximum anti-hepatic liver cancer therapeutic potential were displayed by 2,4-dimethoxy phenyl based bromobenzofuran-oxadiazole **BF-5**. To evaluate the in-depth therapeutic potential and complete mechanism of inhibition of bromobenzofuran-oxadiazoles **BF1-9** against the human hepatocellular carcinoma (HCC), further in vitro studies on other human HCC tissue cell lines would be necessitated in our further studies by utilizing cell proliferation, apoptosis, and autophagy methodologies.

#### 3.2.2. Structure-Activity Relationship of Bromobenzofuran-Oxadiazoles **BF1-9**

The newly synthesized bromobenzofuran-oxadiazoles **BF1-9** contained a variety of substituted acetanilides **2a-I,** which were installed to enhance the lipophilicity of the synthesized scaffolds, and the cytotoxic results were prominently improved as mentioned in Table 2. The SAR studies revealed that the substituted acetanilides depicted arbitrary behavior as bromobenzofuran-oxadiazole compounds **BF-2, BF-5,** and **BF-6,** which have electron withdrawing 3,4-dichloro, 2,4-dimethoxy, and 2-fluoro groups, showed considerably lower cell viability (12.72 ± 2.23%, 10.41 ± 0.66, and 13.08 ± 1.08) and highly remarkable and significant anti-cytotoxic potential against HepG2 cancer cells. Bromobenzofuran-oxadiazoles **BF-3** and **BF-9,** which have electron-donating methyl groups at 2,4- and 3,4-positions on phenyl groups of the anilide, resulted in higher cell viability (33.12 ± 6.15% and 44.69 ± 6.85%) and demonstrated the least anticancer therapeutic efficacies as in presented in Table 2. This study indicated that the bromobenzofuran-oxadiazoles having electron-withdrawing methoxy, fluoro, and chloro groups displayed the best and the most significant anticancer therapeutic potential against HepG2 cancer cells than electron-donating methyl group containing benzofuran-oxadiazoles. The cytotoxic therapeutic potential decreased in the following order: 2,5-dimethoxy > 3,4-dichloro > 2-flouro > 4-flouro > 2-methoxy > 2-Chloro > Pheny > 2,4-dimethyl > 3,4-dimethyl.

It was observed that the second, fourth and fifth positions of phenyl were responsible for the excellent anti-HepG2 cancer activity of bromobenzofuran-oxadizoles **BF1-9,** especially of 2,4-dimethoxy containing a **BF-5** derivative. The introduction of electron withdrawing groups (EWG), such as methoxy, flouro, and chloro on the second, fourth, and fifth positions of phenyl in the bromobenzofuran-oxadiazoles **BF1-9,** enhanced the cytotoxic potential while substituents pattern of the electron donating groups (EDG) on the phenyl of anilide ring decreased the cytotoxic therapeutic potential of bromobenzofuran-oxadiazoles **BF1-9,** as depicted in Figure 4 and Table 2. The effect of different substituents on the phenyl anilide ring displayed a vital role in cytotoxic behavior of bromobenzofuran-oxadiazole structural hybrids **BF1-9**.

### 3.3. Computational Investigations 

#### 3.3.1. Molecular Docking Studies of Bromobenzofuran-Oxadiazoles **BF1-9**

Some of the main targets for liver cancer medication development include several different pathways implicated in crucial cancer-related activities like angiogenesis, cell proliferation, and apoptosis. These pathways include several molecular targets that are reported to have uncontrolled expression rates that help in the invasiveness and proliferation of these cancers in the liver. When it comes to creating anti-cancer drugs, some of the important key signaling molecules/molecular targets of interest include the PI3K/Akt/mTOR, tubulin polymerization, GSK-3β, EGFR, and its related pathways [68,69,70,71,72,73,74].

In our studies, the in vitro investigations showed that three of the synthesized compounds **BF-2**, **BF-5,** and **BF-6** showed good activities against the HepG2 liver cancer cell line and for the prediction of the probable targets of these synthesized compounds. EGFR performs vital roles in the physiology of epithelial cells and is the target of numerous commonly used medicines to treat cancer in clinical practice since it is commonly mutated and/or overexpressed in various types of human malignancies. Several novel benzofuran scaffolds carrying compounds have been reported in the literature to have significant anti-EGFR activities [75,76,77]. Similarly, other important cancer molecular targets and their related pathways like PI3K, Akt, GSK-3β, mTOR, and tubulin polymerization, etc. have also been targeted by these types of compounds, and they exhibit good anti-cancer properties against these molecular targets [78,79,80,81,82,83].

Based on these observations, we performed in silico investigations of the synthesized bromobenzofuran-oxadiazole compounds against these different cancer-related molecular targets. We exploited molecular docking approaches to evaluate the binding affinities and the interactions of three potent compounds **BF-2**, **BF-5,** and **BF-6** against EGFR, PI3K, Akt, GSK-3β, mTOR, and tubulin polymerization, which are important molecular targets in various cancers.

The investigations of these bromobenzofuran-oxadiazole compounds **BF-2**, **BF-5,** and **BF-6** against the EGFR revealed that the bromobenzofuran-oxadiazole compound **BF-5** showed greater efficacy in the in vitro studies; had a binding affinity of −15.17 Kcal/mol with its active site; and made two conventional hydrogen bonds with LYS721, ASP831, and two carbon-hydrogen bonds were observed with the CYS751 and PHE832 of the EGFR active site residues. A single water-assisted hydrogen bond as well as several stabilizing hydrophobic (Pi-sigma, Alkyl, and Pi-Alkyl) interactions were also made by this compound **BF-5** with the active site amino acids of the EGFR enzyme and can be seen in Figure 5.

Similarly, the other two bromobenzofuran-oxadiazole compounds, **BF-2** and **BF-6**, were able to bind with the active site of the EGFR with binding affinities of −14.17 Kcal/mol and −12.59 Kcal/mol, respectively. These two compounds (**BF-2** and **BF-6**) also made significant interactions of different types by engaging the active site residues of EGFR via multiple hydrogen bonds and halogen interactions. These two bromobenzofuran-oxadiazole derivatives showed multiple hydrophobic stabilizing interactions and can be seen in Figure 6.

The docking studies of the standard EGFR Erlotinib inhibitor in these studies showed that Erlotinib bound with the active site of EGFR with a binding affinity of −11.67 Kcal/mol, which suggested that the synthesized novel compounds **BF-2**, **BF-5**, and **BF-6** exhibited higher affinities with EGFR as compared to the standard reference drug Erlotinib. A summary of their binding affinities and their total interactions with the EGFR active site are given in Table 3.

Furthermore, we investigated the binding affinities of these three compounds **BF-2**, **BF-5**, and **BF-6** with the PI3K enzyme which is also implicated in various cancer and is an important drug target for bromobenzofuran-based compounds. The docking investigations of **BF-2**, **BF-5**, and **BF-6** revealed that **BF-2** binds with the PI3K active site with the highest binding affinity of −15.17 Kcal/mol. The interaction analysis of its conformational pose inside the PI3K active site showed that **BF-2** made several different types of stronger hydrogen bonds with the PI3K active site (MET953, ASP836, LYS833, ASP964) along with that of water-assisted hydrogen bonding, hydrophobic interactions (Alkyl and Pi-Alkyl) with ALA885, ILE881, ILE879, ILE963, Pi-Sulfur, Pi-Pi T-shaped, and Amide-Pi interaction with the TYR867, GLY966, and Pi-anion as well van der waals interaction was also observed in the **BF-2**+PI3K complex. It can be seen in Figure 7 in three and two dimensional conformations inside the PI3K active site.

The other two lead compounds **BF-5** and **BF-6,** which showed good efficacy in the in vitro investigations, also exhibited good binding affinities with the PI3K protein active site. Bromobenzofuran-oxadiazole compound **BF-5** was able to bind with the PI3K active site with a binding affinity of −13.17 Kcal/mol and made water-assisted H-bond and two carbon-hydrogen bonds with the LYS833 and ASP964 active pocket residues; other hydrophobic type interactions, including Pi-Alkyl, Pi-Sigma, Pi-Lone Pair, Pi-Anion, and Pi-Sulfur interactions, were also observed between the **BF-5**+PI3K complex. BF-6 showed a binding affinity of −12.90 Kcal/mol and showed the same type of multiple types of hydrogen bonding as well water-assisted H-bonding with LYS833 and ASP964 along with other stabilizing hydrophobic, and Amide-Pi stacked interactions with TYR867, GLY966, ASP836, etc. were also observed their two-dimensional interactive diagrams and are given in Figure 8.

Overall, three bromobenzofuran-oxadiazoles (**BF-2**, **BF-5**, and **BF-6**) showed stronger binding interactions with the PI3K active site while the Idelalisib standard PI3K inhibitor was able to bind to its active site with a binding affinity of −11.42 Kcal/mol, which suggested that these novel **BF-2**, **BF-5**, and **BF-6** compounds could bind more strongly with the PI3K compared to the standard drugs used in our docking studies. The binding affinity energies of these three (**BF-2**, **BF-5**, and **BF-6**) structural motifs arose due to the interactions of these compounds with the PI3K active pocket amino acid residues and are presented in Table 4.

Other than these important cancer molecular targets, the bromobenzofuran-oxadiazole compounds **BF-2**, **BF-5**, and **BF-6** were also evaluated via molecular docking studies against the mTOR, AKT, and Tubulin proteins as well because they are also involved in several cancers, and compounds carrying the benzofuran moiety have been reported several times in the literature as potent inhibitors of these proteins. The molecular docking investigations of bromobenzofuran-oxadiazoles **BF-2**, **BF-5**, and **BF-6** against these proteins revealed that out of these three compounds, **BF-5** and **BF-6** showed higher binding affinities with mTOR than the standard mTOR inhibitor (Torin-2) and showed significantly good interactions with its active site. Similarly, docking investigations against the Tubulin protein showed that **BF-2** and **BF-5** also strongly bind and show good interactions with this protein compared to its standard inhibitor (Colchicine) of Tubulin protein. The binding energies of these compounds against mTOR and Tubulin proteins are given in Table 5, and the two-dimensional diagrams of the best binding compounds against the mTOR and Tubulin proteins are given in Figure 9.

Along with these cancer targets, we also investigated these compounds against the GSK-3β and Akt enzyme, and these studies showed that their binding affinities against the GSK-3β and Akt enzyme were too low compared to the EGFR, PI3K, mTOR, and Tubulin protein molecular targets. Moreover, the computational investigations showed that these compounds bind strongly with the EGFR, PI3K, mTOR, and Tubulin enzymes and have good binding affinities as well as strong interactions, which suggest that these four important cancer-related molecular targets may be the target of these novel bromobenzofuran-oxadiazole compounds.

#### 3.3.2. ADMET and Drug-Likeness Studies of Bromobenzofuran-Oxadiazoles **BF1-9**

According to the ADMET assessment (or pharmacokinetics analyses), these bromobenzofuran-oxadiazoles had acceptable lipophilic (iLogP) qualities, good Log S (ESOL) water solubility values, and good human intestine absorptions, and they were designated as HIA+. They were also non-substrates of the P-gp protein, which controls the efflux of substances and medicines from cells through membrane transport. These substances can readily be bio-transformed inside the liver and then be transferred to the excretory organs for excretion from the body because, according to metabolism studies, the findings showed that they are substrates of the crucial metabolic enzyme (CYP450 3A4). The organic cation transporter (OCTs) protein in the kidneys, which is essential for the body’s removal of foreign chemicals and medications, was also not inhibited by these compounds. The toxicity studies of these molecules also showed that they are non-AMES toxic, non-carcinogenic, and non-interferers of the normal function of the T hERG II ion channel, which controls cardiac action potential repolarization. The complete profile of its ADME&T investigations is presented in Table 6 while its structures along with the graphical pharmacokinetic profiles can be seen in Figure 10.

According to the drug-likeness investigations that involve the identification of the physicochemical attributes and medicinal chemistry of compounds, these bromobenzofuran structural hybrids **BF-2, BF-5**, and **BF-6** possessed good topological surface area (TPSA), acceptable molecular weight values, and good synthetic accessibility scores, as shown in Table 7. These substances adhered to all drug-likeness guidelines, including the Lipinski and Pfizer rules. These substances had good bioavailability scores (greater than 0.10), did not exhibit any PAINS alarms, and obeyed the Golden Triangle rule. The three lead bromobenzofuran-oxadiazole compounds **BF-2, BF-5**, and **BF-6** showed noticeably good ADMET and drug-likeness qualities. On the basis of all the investigations analysis, these bromobenzofuran-oxadiazoles leads can safely be developed as potential pharmaceuticals. To evaluate the in depth therapeutic potential and complete mechanism of inhibition of bromobenzofuran-oxadiazoles **BF1-9** against the human hepatocellular carcinoma (HCC), further in vitro studies on other Human HCC tissue cell lines would be necessitated in our further studies by utilizing cell proliferation, apoptosis, and autophagy methodologies.

#### 3.3.3. Molecular Dynamic Simulations of Bromobenzofuran-Oxadiazoles **BF1-9**

The dynamics assessment of the most bioactive docked complexes PI3K+**BF-2** and EGFR+**BF-5** were done through a molecular dynamics simulation technique. The simulation trajectories were studied for the structural stability of bromobenzofuran-oxadiazoles with receptors via the root mean square deviation (RMSD), root mean square fluctuation (RMSF), and radius of gyration (RoG). All of these analyses were done based on carbon alpha atoms. Generally, all the analyses predicted the very stable formation of stable complexes. RMSD plots (A part of Figure 11) reported the very stable behavior of the PI3K+**BF-2** complex throughout the simulation time while the EGFR+**BF-5** complex initially experienced some deviation in the first 35 ns. The RMSD of both system maxima touches 4 angstroms. This was also complemented with an RMSF analysis, which complemented the RMSD findings and found the receptors residues in the presence of compounds very stable (B part of Figure 11). The PI3K+**BF-2** complex receptor reported the C-terminal being very flexible compared to the rest of the enzyme structure. The RoG analysis was another confirmation of the RMSD findings and unveiled the systems to have compact nature in the compound’s presence.

Binding Free Energy Analysis Further validation of the docking and simulation findings was accomplished using MMGBSA and MMPBSA methods. Both the methods are now frequently used in modern drug discovery as they use modest computational speed and correlate well with the experimental data. The estimated binding free energy results are tabulated in Table 8. As can be seen, both complexes in MMGBSA and MMPBSA are very much stable, as can be understood by −38.44 kcal/mol (MMGBSA) and −42.55 kcal/mol (MMPBSA) for the EGFR+**BF-5** complex and −39.54 kcal/mol (MMGBSA) and −45.13 kcal/mol (MMPBSA) for the PI3K+**BF-2** complex. 

The results indicated stable binding conformation of the compounds with the receptors and formed strong intermolecular interactions. The van der Waals and electrostatic energies played a vital role in the complex’s stability while a negative contribution was seen from the polar energy component.

#### 3.3.4. DFT Studies of Bromobenzofuran-Oxadiazoles **BF1-9**

The energy computations of the three bioactive bromobenzofuran-oxadiazole analogues **BF-2, BF-5**, and **BF-6** were undertaken with the Gaussian calculation setups used in the optimization. From the DFT energy computation outcomes, the total energy, the HOMO energy, and the LUMO energy were calculated. By using the HOMO and LUMO energy figures, the other related parameters were calculated with the respective theorems, as presented in Table 9.

Both the HOMO and LUMO play an important role in estimating the electrical properties and chemical affinities of the bromobenzofuran-oxadiazole **BF-2, BF-5**, and **BF-6** compounds. The HOMO depicts the electron donors. On the other hand, the LUMO depicts the electron acceptors [87,88]. The HOMO energy value of compound **BF-5** was found to be the highest followed by the compounds **BF-6** and **BF-2,** respectively. Therefore, compound BF-**5** is expected to have the highest tendency to give electrons easily as depicted in Table 8. The HOMO–LUMO energy gap (∆E) exhibits the chemical stability of compounds. A higher energy gap for a molecule implies higher chemical stability [89]. In the DFT study, compound **BF-2** had the highest energy gap among the three compounds. Hence, compound **BF-2** is expected to have the highest chemical stability. Global hardness depicts the resistance of an atom to electron transfer. Here, compound **BF-2** had the highest global hardness. From these outcomes of DFT analysis, it Is inferred that bromobenzofuran-oxadiazole compound **BF-2** is the least reactive and has the highest chemical stability among the three bioactive **BF-2, BF-5**, and **BF-6** compounds [90]. The concentration of tubes for compound **BF-2** was around the benzofuran ring. On the other hand, for compound **BF-5**, the tubes were concentrated around the dimethoxyphenyl ring. The LUMO tubes for compound **BF-6** were concentrated around the benzofuran and the oxadiazole rings. However, its HOMO tubes were concentrated not only around the two heterocyclic structures but also around the fluoro phenyl ring, as presented in Figure 12. The DFT study revealed that the 2,5-dimethoxy-based benzofuran-oxadiazole **BF-5** can be the lead anti-HepG2 liver cancer structural motifs.

## 4. Conclusions

In the present work, nine *S*-alkylated amide-linked bromobenzofuran-oxadiazole scaffolds **BF1-9** were achieved by employing the phase transfer catalyst BTEAC under the ultrasonic-assisted synthetic conditions in good yields (75–88%). These synthesized bromobenzofuran oxadiazoles **BF1-9** were evaluated for cytotoxic potential against human liver cancer cell line HepG2 and obtained promising results with cell viabilities of 10.41 ± 0.66% to 44.69 ± 6.85%. The least active 3,4-dimethyl containing benzofuran-oxadiazole **BF-9** showed maximum cell viability potential (44.69 ± 6.85%), and the most bioactive 3,4-dimethoxy based benzofuran-oxadiazole demonstrated the minimum cell viability efficacy (10.41 ± 0.66%) and excellent anti-HepG2 cytotoxic activity. The compounds 3,4-Dichloro-based **BF-2** (12.72 ± 2.23% cell viability) and 2-fluoro-based **BF-6** (13.08 ± 1.08% cell viability) displayed good cytotoxic potential. The cytotoxic therapeutic potential of **BF1-9** derivatives decreased in the following order: 2,5-dimethoxy > 3,4-dichloro > 2-flouro > 4-flouro > 2-methoxy > 2-chloro > pheny > 2,4-dimethyl > 3,4-dimethyl. SAR revealed that the second and fourth positions of phenyl are responsible for the excellent anti-HepG2 cancer activity of benzofuran-oxadiazoles **BF1-9,** especially of 2,5-dimethoxy containing **BF-5** derivative. The EWGs such as methoxy, fluoro, and chloro on the second, fourth, and fifth positions of phenyl in the anilide ring of benzofuran-oxadiazoles **BF1-9** could enhance the cytotoxic potential while substituents pattern of the electron-donating groups (EDG) on the phenyl of anilide ring decreased the cytotoxic therapeutic potential of bromobenzofuran-oxadiazoles **BF1-9**. Furthermore, the in silico investigations for the identification of its probable mechanism of anti-cancer action revealed that these compounds showed good binding affinities and stable associations with the EGFR, PI3K, mTOR, and Tubulin polymerzation enzymes. This study indicates that these four key enzymes may be the probable molecular targets for these compounds **BF1-9**. Based on the significantly good binding affinities and associations, these in silico studies further validate the anti-HepG-2 cancer potential as witnessed in the in vitro evaluations. The novel **BF1-9** analogues did not demostrate significant in silico results against the GSK-3β and AKT enzymes. The ADMET assessment demonstrated that novel bromobenzofuran-oxadiazoles **BF1-9** members have a high degree of drug-likeness profile. The DFT studies also revealed that bromobenzofuran-oxadiazole compound **BF-2** had the highest chemical stability while on the other hand, bromobenzofuran-oxadiazole structural motif **BF-5** was found to have the highest tendency to give its electrons easily as compared to **BF-2** and **BF-6**. All the in vitro and in silico findings of the present research work against anti-HepG-2 cancer cell line and four key enzymes EGFR, PI3K, mTOR, and Tubulin polymerization conclude that the 2,5-dimethoxy based bromobenzofuran-oxadiazole **BF-5** could be the potential lead anti-HepG2 cancer agent.

## Data Availability

Data is available in the manuscript.

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
