# Peer review of "BTEAC Catalyzed Ultrasonic-Assisted Synthesis of Bromobenzofuran-Oxadiazoles: Unravelling Anti-HepG-2 Cancer Therapeutic Potential through In Vitro and In Silico Studies"

_ijms, 2023, doi:10.3390/ijms24033008_

Round 1
Reviewer 1 Report
1. For the BTEAC catalyzed ultrasonic-assisted synthesis of bromoben-zofuran- oxadiazoles, this study was designed to evaluate the anti-hepatocellular carcinoma therapeutic potential. However, only HepG2 liver cancer cell line was used. It is not very enough and there is a long way to apply it in HCC treatment. Other HCC cell lines are suggested to be used, or the title should be modified.
2. The writing is difficult and verbose. For example: (1) Three Figures were used in the Introduction; and (2) For the major cause of hepatocellular carcinoma (HCC), nine references were cited (Ref 5-13). Some of them can be simplified.
3. For the anti-hepatocellular carcinoma therapeutic potential, In-vitro and In-silico studies are not enough. Human HCC tissue specimen are suggested to be used, or the title and abstract should be modified.
4. The MTT assay was used to evaluate the anti-hepatocellular therapeutic potential of the afforded bromobenzofuranoxadiazoles BF1-9. Multiple mechanisms are involved in the development and progression of HCC, for example cell proliferation, apoptosis and autophagy. For the anti-hepatocellular therapeutic potential, only MTT assay was used and cell proliferation was evaluated in this study. It was not very enough.
5. As descried by the authors, some of the important key signaling molecules/ molecular targets of interest include the PI3K/Akt/mTOR, Wnt/β-catenin, GSK-3β, EGFR and its related pathways. However, only EGFR, PI3K, and GSK-3β were examined for these molecular pathways in this study. It is difficult to reach the conclusion of “Dual Inhibition of EGFR and PI3K Signaling Pathways”.
6. For the References, 110 references were cited and some of them could be deleted.
Author Response
Dear Worthy Editor & Reviewer-1
International Journal of Molecular Sciences
Subject:BTEAC Catalyzed Ultrasonic-Assisted Synthesis of Bromobenzofuran-Oxadiazoles: Unravelling Anti-HepG-2 Cancer Therapeutic Potential through In-vitro and In-silico Studies (Manuscript ID: ijms-2130354)
Dear Sir/Miss,
Thank you very much for peer reviewing our manuscript and we appreciate your complimentary recommendations as your comments have helped us significantly to improve the manuscript. We have carefully scrutinized the suggestions mentioned by our worthy reviewers and in accordance of reviewer’s comments, we have revised the manuscript.
In general, all the recommendations and suggestions have been addressed and incorporated in the manuscript which include following.
POINT 1:For the BTEAC catalyzed ultrasonic-assisted synthesis of bromoben-zofuran- oxadiazoles, this study was designed to evaluate the anti-hepatocellular carcinoma therapeutic potential. However, only HepG2 liver cancer cell line was used. It is not very enough and there is a long way to apply it in HCC treatment. Other HCC cell lines are suggested to be used, or the title should be modified.
RESPONSE 1:According to Worthy reviewer suggestion, title has been modified as:
BTEAC Catalyzed Ultrasonic-Assisted Synthesis of Bromobenzofuran-Oxadiazoles: Unravelling Anti-HepG-2 Cancer Therapeutic Potential through In-vitro and In-silico Studies.
POINT 2:The writing is difficult and verbose. For example: (1) Three Figures were used in the Introduction; and (2) For the major cause of hepatocellular carcinoma (HCC), nine references were cited (Ref 5-13). Some of them can be simplified.
RESPONSE 2:Introduction language is improved and references have been reduced (References 5-9) as per instruction of worthy reviewer. Changes are highlighted in green color.
POINT 3:For the anti-hepatocellular carcinoma therapeutic potential, In-vitro and In-silico studies are not enough. Human HCC tissue specimen are suggested to be used, or the title and abstract should be modified.
RESPONSE 3:Dear reviewer, at this stage the further in-vitro analysis is not possible but further in-silico studies have been carried out against mTOR, AKT and tubulin polymerization enzymes and both the abstract and title have been modified accordingly.
POINT 4:The MTT assay was used to evaluate the anti-hepatocellular therapeutic potential of the afforded bromobenzofuranoxadiazoles BF1-9. Multiple mechanisms are involved in the development and progression of HCC, for example cell proliferation, apoptosis and autophagy. For the anti-hepatocellular therapeutic potential, only MTT assay was used and cell proliferation was evaluated in this study. It was not very enough.
RESPONSE 4:Respected reviewer, we acknowledge your opinion but at this stage furthercell proliferation, apoptosis and autophagy studies are not possible, we will apply your valuable suggestion in our future research work. However, further in-silico modelling studies have been carried out to determine inhibitory therapeutic potential against AKT, mTOR and Tubulin Polymerization along with EGFR, PI3K, and GSK-3β signalling pathways to deduce conclusion.
POINT 5:As described by the authors, some of the important key signaling molecules/ molecular targets of interest include the PI3K/Akt/mTOR, Wnt/β-catenin, GSK-3β, EGFR and its related pathways. However, only EGFR, PI3K, and GSK-3β were examined for these molecular pathways in this study. It is difficult to reach the conclusion of “Dual Inhibition of EGFR and PI3K Signaling Pathways”.
RESPONSE 5:Respected reviewer, these three pathways should be enough (pls see references below) along with in-vitro data for confirmation of anti-HepG-2 potential of furan-oxadiazole compounds but in routine publication one or two and maximum three enzymes targets (pls see references below) are selected but we carried out in-silico studies against six (6) cancer target enzymes. However, as mentioned in the comment 4 response, we have performed further molecular docking techniques against all important and key signalling pathways/molecular targets such as AKT, mTOR and Tubulin Polymerization to confirm the inhibitory potential of furan-oxadiazole structural motifs against ant-HepG-2 cancer cell line to deduce conclusion as per your suggestion. These compounds showed excellent docking scores against mTOR and tubulin polymerization respectively. The furan-oxadiazoles showed weak binding affinities with AKT enzyme similar to GSK-3β as mentioned in manuscript.
1-Mphahlele, M.J.; Maluleka, M.M.; Parbhoo, N.; Malindisa, S.T. Synthesis, Evaluation for Cytotoxicity and Molecular Docking Studies of Benzo[c]Furan-Chalcones for Potential to Inhibit Tubulin Polymerization and/or EGFR-Tyrosine Kinase Phosphorylation. Int. J. Mol. Sci. 2018, 19,2552. doi:10.3390/ijms19092552.
2- Tsai, K.H.; Hsien, H.H.; Chen, L.M.; Ting, W.J.; Yang, Y.S.; Kuo, C.H.; Tsai, C.H.; Tsai, F.J.; Tsai, H.J.; Huang, C.Y. Rhubarb Inhibits Hepatocellular Carcinoma Cell Metastasis via GSK-3-β Activation to Enhance Protein Degradation and Attenuate Nuclear Translocation of β-Catenin. Food Chem. 2013, 138, 278–285. doi:10.1016/j.foodchem.2012.10.038
3-Zhu, M.; Li, W.; Lu, Y.; Dong, X.; Lin, B.; Chen, Y.; Zhang, X.; Guo, J.; Li, M. HBx Drives Alpha Fetoprotein Expression to Promote Initiation of Liver Cancer Stem Cells through Activating PI3K/AKT Signal Pathway. Int. J. Cancer. 2017, 140, 1346–1355. doi:10.1002/ijc.30553
4-El-Khouly, O.A.; Henen, M.A.; El-Sayed, M.A.A.; Shabaan, M.I.; El-Messery, S.M. Synthesis, Anticancer and Antimicrobial Evaluation of New Benzofuran Based Derivatives: PI3K Inhibition, Quorum Sensing and Molecular Modeling Study. Bioorganic Med. Chem. 2021, 31, 115976. doi:10.1016/j.bmc.2020.115976.
5-Sankhe, N. M.; Durgashivaprasad, E.; Kutty, N. G.; Rao, J. V.; Narayanan, K.; Kumar, N.; Jain, P.; Udupa, N.; Palanimuthu, V. R. Novel 2,5-disubstituted-1,3,4-oxadiazole derivatives induce apoptosis in HepG2 cells through p53 mediated intrinsic pathway.Arab. J. Chem. 2015. https://doi.org/10.1016/j.arabjc.2015.04.03
6- Sun, E.J.; Wankell, M.; Palamuthusingam, P.; McFarlane, C.; Hebbard, L. Targeting the PI3K/Akt/mTOR Pathway in Hepatocellular Carcinoma. Biomedicines 2021, 9, 1639. https://doi.org/10.3390/biomedicines9111639
7-Luo, X., Cao, M., Gao, F. et al. YTHDF1 promotes hepatocellular carcinoma progression via activating PI3K/AKT/mTOR signaling pathway and inducing epithelial-mesenchymal transition. ExpHematolOncol 10, 35 (2021). https://doi.org/10.1186/s40164-021-00227-0
8- Li A, Zhang R, Zhang Y, Liu X, Wang R, Liu J, Liu X, Xie Y, Cao W, Xu R, Ma Y, Cai W, Wu B, Cai S, Tang X. BEZ235 increases sorafenib inhibition of hepatocellular carcinoma cells by suppressing the PI3K/AKT/mTOR pathway. Am J Transl Res. 2019 Sep 15;11(9):5573-5585
9-Moustafa, M.; Dähling, K.-K.; Günther, A.; Riebandt, L.; Smit, D.J.; Riecken, K.; Schröder, C.; Zhuang, R.; Krech, T.; Kriegs, M.; Fehse, B.; Izbicki, J.R.; Fischer, L.; Nashan, B.; Li, J.; Jücker, M. Combined Targeting of AKT and mTOR Inhibits Tumor Formation of EpCAM+ and CD90+ Human Hepatocellular Carcinoma Cells in an Orthotopic Mouse Model. Cancers 2022, 14, 1882. https://doi.org/10.3390/cancers14081882
10-Gaisina, I.N.; Gallier, F.; Ougolkov, A. V.; Kim, K.H.; Kurome, T.; Guo, S.; Holzle, D.; Luchini, D.N.; Blond, S.Y.; Billadeau, D.D.; et al. From a Natural Product Lead to the Identification of Potent and Selective Benzofuran-3-Yl-(Indol-3-Yl)Maleimides as Glycogen Synthase Kinase 3βinhibitors That Suppress Proliferation and Survival of Pancreatic Cancer Cells. J. Med. Chem. 2009, 52, 1853–1863. doi:10.1021/jm801317h
11- Saitoh, M.; Kunitomo, J.; Kimura, E.; Iwashita, H.; Uno, Y.; Onishi, T.; Uchiyama, N.; Kawamoto, T.; Tanaka, T.; Mol, C.D.; et al. 2-{3-[4-(Alkylsulfinyl)Phenyl]-1-Benzofuran-5-Yl}-5-Methyl-1,3,4-Oxadiazole Derivatives as Novel Inhibitors of Glycogen Synthase Kinase-3β with Good Brain Permeability. J. Med. Chem. 2009, 52, 6270–6286. doi:10.1021/jm900647e
12- Abbas, H.A.S.; Abd El-Karim, S.S. Design, Synthesis and Anticervical Cancer Activity of New Benzofuran–Pyrazol-Hydrazono- Thiazolidin-4-One Hybrids as Potential EGFR Inhibitors and Apoptosis Inducing Agents. Bioorg. Chem. 2019, 89. doi:10.1016/j.bioorg.2019.103035
POINT 6: For the References, 110 references were cited and some of them could be deleted.
RESPONSE 6:Dear worthy reviewer, references have been reduced to 90.
We hope that revised manuscript would be satisfying for all requirements and will be suitable for consideration for publication.
Kind Regards
Corresponding Author
Reviewer 2 Report
The paper is very articulated, but well done. In fact, it summarizes many characteristics of the BF1-9 molecules. Given the complexity and the numerous experiments, for a better reading I ask for the modifications that must lead to the conclusion being better deduced.
- For each highlighted feature, highlight in the table or figure which is the best molecule (bold in tab, arrows in figure)
-To better compare data from different experiments, find a way to quantify how far apart they are. For example, can you calculate the % difference of all the data compared to the best one found for a single experiment? Or propose an alternative method.
- Arguing better the sentence in the most important conclusions of the work: "On the basis of in vitro and in silico findings in the present research study, the 2,4-dimethoxy based benzofuran-oxadiazole BF-5 will be the best potential lead anti-hepatic cancer agent".
Author Response
Dear Worthy Editor &Reviewer-2
International Journal of Molecular Sciences
Subject:BTEAC Catalyzed Ultrasonic-Assisted Synthesis of Bromobenzofuran-Oxadiazoles: Evaluation of Anti-Hepatocellular Carcinoma Therapeutic Potential through Dual Inhibition of EGFR and PI3K Signaling Pathways: Mechanistic Approach through In-vitro and In-silico(Manuscript ID:ijms-2130354)
Dear Sir/Miss,
Thank you very much for peer reviewing our manuscript and we appreciate your complimentary recommendations as your comments have helped us significantly to improve the manuscript. We have carefully scrutinized the suggestions mentioned by our worthy reviewers and in accordance of reviewer’s comments, we have revised the manuscript.
In general, all the recommendations and suggestions have been addressed and incorporated in the manuscript which include following.
POINT 1:For each highlighted feature, highlight in the table or figure which is the best molecule (bold in tab, arrows in figure)
RESPONSE 1:Worthy reviewer, changes have been made as per your suggestion.
POINT 2:To better compare data from different experiments, find a way to quantify how far apart they are. For example, can you calculate the % difference of all the data compared to the best one found for a single experiment? Or propose an alternative method.
RESPONSE 2:Respected reviewer, the percentage yield range of previous reported work were little low (53-79%) ref [39], however, with our current work it is up to 88% clearly indicating upper hand of this methodology. Important synthetic and biological results have been highlighted bold in tables and figures.
POINT 3:Arguing better the sentence in the most important conclusions of the work: "On the basis of in vitro and in silico findings in the present research study, the 2,4-dimethoxy based benzofuran-oxadiazole BF-5 will be the best potential lead anti-hepatic cancer agent".
RESPONSE 3:Worthy reviewer, your suggestion has been incorporated in the conclusion part of the manuscript,
We hope that revised manuscript would be satisfying for all requirements and will be suitable for consideration for publication.
Kind Regards
Corresponding Author
Round 2
Reviewer 1 Report
Dear authors,
I appreciate the work that you did. However, I am not very satisfied with the revision.
1. For point 3 and point 4, please include the limitations in the DISCUSSION.
2. I suggested that your manuscript should undergo extensive English revisions. But the revision is not obvious.
Author Response
Dear Worthy Editor & Reviewer-1
International Journal of Molecular Sciences
Subject: BTEAC Catalyzed Ultrasonic-Assisted Synthesis of Bromobenzofuran-Oxadiazoles: Unravelling Anti-HepG-2 Cancer Therapeutic Potential through In-vitro and In-silico Studies (Manuscript ID: ijms-2130354)
Dear Sir/Miss,
Thank you very much for peer reviewing our manuscript and we appreciate your complimentary recommendations as your comments have helped us significantly to improve the manuscript. We have carefully scrutinized the suggestions mentioned by our worthy reviewer and in accordance of reviewer’s comments, we have revised the manuscript and 2nd round revisions are highlighted in yellow color.
We hope that revised manuscript would be satisfying for all requirements and will be suitable for consideration for publication.
Kind Regards
Corresponding Authors
